# META-LEARNING VIA CLASSIFIER(-FREE) GUIDANCE

## ABSTRACT

State-of-the-art meta-learning techniques do not optimize for zero-shot adaptation to unseen tasks, a setting in which humans excel. On the contrary, meta-learning algorithms learn hyperparameters and weight initializations that explicitly optimize for few-shot learning performance. In this work, we take inspiration from recent advances in generative modeling and language-conditioned image synthesis to propose meta-learning techniques that use natural language guidance to achieve higher zero-shot performance compared to the state-of-the-art. We do so by recasting the meta-learning problem as a multi-modal generative modeling problem: given a task, we consider its adapted neural network weights and its natural language description as equivalent multi-modal task representations. We first train an unconditional generative hypernetwork model to produce neural network weights; then we train a second "guidance" model that, given a natural language task description, traverses the hypernetwork latent space to find high-performance task-adapted weights in a zero-shot manner. We explore two alternative approaches for latent space guidance: "HyperCLIP"-based classifier guidance and a conditional Hypernetwork Latent Diffusion Model ("HyperLDM"), which we show to benefit from the classifier-free guidance technique common in image generation. Finally, we demonstrate that our approaches outperform existing meta-learning methods with zero-shot learning experiments on our Meta-VQA dataset, which we specifically constructed to reflect the multi-modal meta-learning setting.

## 1 INTRODUCTION

State-of-the-art machine learning algorithms often lack the ability to quickly generalize in a sample efficient manner to new unseen tasks. In contrast, humans show remarkable capabilities in leveraging previous knowledge for learning a new task from just a few examples. Often, not even a single example is needed, as all relevant task information can be conveyed in the form of natural language instructions. Indeed, humans can solve novel tasks when prompted from a variety of different interaction modalities such as visual task observations or natural language prompts. In this work we aim to improve forward generalization of deep networks by developing new models that are capable of a similar multi-modal task inference. In particular, we present two different approaches that utilize text-based task descriptors for rapid zero-shot adaptation to new tasks.

The development of deep learning models that quickly adapt and generalize to unseen tasks is the focus of the field of meta-learning. A recent trend stemming from the success of Model-Agnostic Meta-Learning (Finn et al., 2017, MAML) describes meta-learning as a bi-level optimization problem: an outer loop meta-model is trained with the goal of improving the few-shot performance of a base model when fine-tuned on a variety of related tasks. MAML was specifically introduced as a gradient-based method to find an unconditional network initialization with good few-shot performance over an entire task universe. Recent progress in large scale deep learning is however challenging this explicit meta-learning framework. Large models trained on huge, rich, and diverse data sets have been shown to possess surprisingly good few-shot learning capabilities (Brown et al., 2020). In particular, large scale pre-training and fine-tuning often outperforms explicit meta-learning procedures (Mandi et al., 2022).

These developments prompted us to develop alternative methods for explicit meta-learning which natively benefit from rich and multi-modal data. Inspired by recent advances in conditional image generation (Ramesh et al., 2022; Rombach et al., 2022), we reframe meta-learning as a multi-modal generative modeling problem such that, given a task, its adapted neural network weights and its

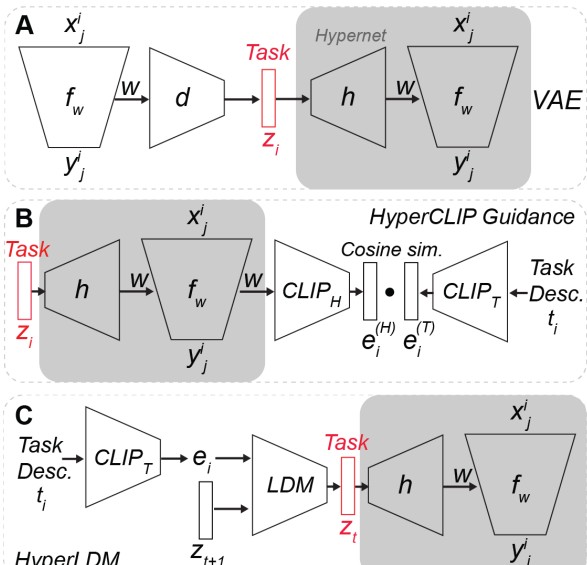

Figure 1: Schematic of the three main components of our proposed meta-learning approach. **A.** An unconditional variational autoencoder (VAE) models the latent space of adapted network weights $W$. Its generator hypernetwork $h$ (highlighted in gray) can be re-used in the conditional setting with our guidance techniques. **B.** Our HyperCLIP encoder $\text{CLIP}_H$ is contrastively trained to map network weights $W$ to the space of CLIP embeddings $e_i$. Then, given a new task with descriptor $t_i$, we can use CLIP guidance to find a VAE latent vector $z_i$ with embedding $e_i^{(H)}$ that has a high cosine similarity to a given task embedding $e_i^{(T)}$. **C.** Alternatively, our Hypernetwork Latent Diffusion Model (HyperLDM) learns, conditional on the task embedding $e_i$, to iteratively denoise a VAE latent vector $z_i^T, \ldots, z_i^0$ over $T$ iterations.

natural language description are considered equivalent multi-modal task representations. What we show is that popular techniques for the image domain, such as CLIP-based guidance (Gal et al., 2021; Patashnik et al., 2021), denoising diffusion models (Ho et al., 2020), and classifier-free guidance (Dhariwal & Nichol, 2021; Ho & Salimans, 2021; Nichol et al., 2022) can be repurposed for the meta-learning setting to generate adapted neural network weights instead of images.

Specifically, we approach the generation of neural network weights in two separate phases. In the *unconditional pre-training* phase, we train a generative hypernetwork (Ha et al., 2016) to map from its latent space to the weight space of a base model (Figure 1.A). In the *guidance* phase, we learn language-conditioned models that can be used to traverse the hypernetwork latent space and find zero-shot adapted weights with high performance on our task (Figure 1.B and 1.C).

## 1.1 OUR CONTRIBUTIONS

We summarise our contributions as follows:

- We recast meta-learning as a multi-modal generative modeling problem using hypernetworks, providing a theoretical framework for the application of recent research on image synthesis to the field of meta-learning.

- We propose **HyperCLIP**, a contrastive learning method equivalent to Contrastive Language-Image Pre-training (CLIP) (Radford et al., 2021), producing CLIP embeddings of fine-tuned neural network weights. We can then use HyperCLIP to: 1) perform CLIP-guidance in the latent space of a hypernetwork model (Figure 1.B); and 2) perform task inference on fine-tuned models without given task descriptions.

- We propose Hypernetwork Latent Diffusion Models (**HyperLDM**) as an alternative to HyperCLIP guidance for task-conditional network generation within the latent space of a hypernetwork model (Figure 1.C). We show how combining this approach with classifier-free guidance (Ho & Salimans, 2021) improves the performance of generated networks.

- We demonstrate the soundness of our methods on **Meta-VQA**, our modification of the Visual-Question-Answering *VQA v2.0* dataset (Goyal et al., 2017) that interprets questions as task descriptions. We show how our guidance methods outperform traditional multi-task and meta-learning techniques for zero-shot learning on this dataset.

## 2 META-LEARNING WITH MULTI-MODAL TASK EMBEDDINGS

The setting we investigate is similar to the classic meta-learning framework, where we operate within a distribution of tasks $\mathcal{T}_i \sim p(\mathcal{T})$, each associated with a loss function $\mathcal{L}_{\mathcal{T}_i}$. Using a set of training tasks drawn from this distribution, our goal is to train a model such that it generally performs well on new unseen tasks drawn from $p(\mathcal{T})$.

### 2.1 MODEL-AGNOSTIC META-LEARNING

We present here a slightly altered formulation of MAML (Finn et al., 2017) introduced in (Zintgraf et al., 2019), whereby the parameters of the model $g$ are partitioned into two parts: context parameters $\phi$ that are adapted on individual tasks, and shared parameters $\theta$ that are meta-trained and shared across tasks. MAML and its variants focus on the few-shot setting, which aims to learn an initialization for these parameters such that the model $g(\cdot, \theta, \phi)$ generalizes well on new tasks after fine-tuning $\phi$ on a few data points from that task. To train such a model, the data from each task $\mathcal{T}_i$ is split during training into a support set $D_i^s$ and a query set $D_i^q$. The MAML objective aims to optimize the validation score evaluated on the query set when fine-tuning $\phi$ on the support set, *e.g.*, consider the following optimization problem:

$$\min_{\theta,\phi} \mathbb{E}_{\mathcal{T}_i \sim p(\mathcal{T})} \left[ \frac{1}{|D_i^q|} \sum_{(x,y) \in D_i^q} \mathcal{L}_{\mathcal{T}_i} \left( g(x, \theta, \mathcal{A}_{\mathcal{T}_i}(D_i^s, \theta, \phi)), y \right) \right], \tag{1}$$

where $\mathcal{A}_{\mathcal{T}_i}$ is some differentiable algorithm, typically implementing a variant of few-step gradient descent on the loss computed on the support set, *e.g.*, in the case of one-step gradient descent:

$$\mathcal{A}_{\mathcal{T}_i}(D_i^s, \theta, \phi) = \phi - \eta \frac{1}{|D_i^s|} \sum_{(x',y') \in D_i^s} \nabla_\phi \mathcal{L}_{\mathcal{T}_i}(g(x', \theta, \phi), y') \tag{2}$$

with some learning rate $\eta$. The objective from Eq. 1 is itself solved with gradient descent, by iteratively optimizing the parameters $\phi$ in the inner loop on the support set of a sampled task, and updating $\theta$ and the initialization of $\phi$ with their gradient with respect to the entire inner loop training process, averaged over batches of tasks. Note that the original formulation of MAML considers $\theta = \emptyset$.

### 2.2 NATURAL LANGUAGE TASK EMBEDDINGS

In this work, we assume to have access to an additional high-level context embedding $e_i$ for each task $\mathcal{T}_i$. In practice, such embeddings can come from a natural language description $t_i$ of the task, which can be encoded into task embeddings using pre-trained language models. While MAML is a powerful algorithm for finding universal initializations, the model naturally requires some data at test time to adapt its unconditioned network parameters. In contrast, by incorporating task embeddings, we aim to improve performance on a new task in the no-data (zero-shot), or very low data regime.

A simple way to incorporate task embeddings into our model during training is by augmenting the input of the network, concatenating such input with the task embedding during the forward pass. Instead, we consider the use of hypernetworks (Ha et al., 2016), a network that generates the weights of another network given a conditioning input. Concretely, given an unconditioned neural network $f$ parametrized by a weight vector $W$, we reparametrize the model by introducing a hypernetwork $h$. The hypernetwork $h$ is parametrized by $\theta$, which generates a task-conditioned weight, $h(e_i, \theta) = W_i$. The model is then defined as $f(\cdot, h(e_i, \theta))$.

We can thus rewrite the MAML objective with respect to the hypernetwork weight $\theta$ as

$$\min_{\theta} \mathbb{E}_{\mathcal{T}_i \sim p(\mathcal{T})} \left[ \frac{1}{|D_i^q|} \sum_{(x,y) \in D_i^q} \mathcal{L}_{\mathcal{T}_i} \left( f(x, h(\mathcal{A}_{\mathcal{T}_i}(D_i^s, e_i, \theta), \theta)), y \right) \right], \tag{3}$$

when $\mathcal{A}_{\mathcal{T}_i}(D_i^s, e_i, \theta) = e_i$, we recover the classic multi-task objective of a hypernetwork optimizing for zero-shot performance. When $\mathcal{A}_{\mathcal{T}_i}$ is instead the gradient descent algorithm on $e_i$, the objective aligns with the few-shot performance of $h$ when adapting the embedding initialized at $e_i$.

## 3 HYPERNETWORKS AS GENERATIVE MODELS

A rich literature exists on hypernetworks interpreted as generative models of base network weights (see Section 7). Our work builds upon this interpretation to adapt multi-modal generative modeling techniques to the meta-learning domain.

In generative modeling, we aim to learn the distribution $p(x)$ over a high dimensional data domain $\mathcal{X}$, such as images, given samples from the distribution. Since classic maximum-likelihood learning is insufficient to do so, we must resort to techniques such as variational inference, adversarial learning, or diffusion models. It seems apparent that, in meta-learning, the modeling of a distribution of diverse high-dimensional base network weights $W$ may be considered in analogous terms. In the Bayesian setting, this distribution is made explicit as we seek to model the posterior $p(W|D)$ given data $D$, but the framework is still useful even when no explicit posterior distribution is assumed, as it is the case for deep ensembles. In the present work, we indeed avoid explicit Bayesian inference: given training tasks, we consider their respective fine-tuned network weights $W_i$ as training samples for our generative model of network weights.

The fundamental building block of our unconditional generative model is the hypernetwork $h(z, \theta) = W$ that we can train in two ways: **1)** We define a Hypernetwork VAE (HVAE) as in Figure 1.A, which, given samples of fine-tuned base network weights $W_i$, learns a low-dimensional normally distributed latent representation $z$. The encoder $d(W, \omega) = (\mu_z, \Sigma_z)$ with parameters $\omega$ maps base network weights to means and variances used to sample a latent vector $z$, while the decoder (or generator) is a classic hypernetwork $h(z, \theta) = W$ which reconstructs the network weights from the latent vector. **2)** Using MAML, we learn both an embedding $z$ and hypernetwork weights $\theta$ such that, when fine-tuning only the embedding $z$ on each task $\mathcal{T}_i$, we obtain high-performing base networks with weights $W_i = h(z_i, \theta)$. Concretely, we optimize $\theta$ and the initialization of $z$ following the objective in Eq. 1 where $z$ takes the role of the task-specific parameter $\phi$.

## 4 HYPERCLIP: TRAINING A CLIP ENCODER FOR THE "META-LEARNING MODALITY"

To define our new meta-learning approach, we first borrow from the field of multi-modal contrastive learning. More specifically, we build on top of Contrastive Language-Image Pre-training (CLIP) (Radford et al., 2021), a popular method for joint learning of language and image embeddings with applications to zero-shot and few-shot classification.

In the original CLIP formulation, separate text and image encoders are trained such that, given a bi-modal sample $(x_i, t_i)$ of an image and its corresponding language caption, their representations ($\text{CLIP}_I(x_i) = e_i^{(I)}$ and $\text{CLIP}_T(t_i) = e_i^{(T)}$) are aligned across modalities. Specifically, the formulation maximizes the cosine similarity $e_i^{(I)\top} e_j^{(T)} / \|e_i^{(I)}\| \|e_j^{(T)}\|$ for pair-wise matches ($i = j$) and minimizes the cosine similarity for non-matches ($i \neq j$). Beyond the original language-image setting, the CLIP approach can be easily adapted to include additional modalities, aligning the representation of more than two encoders at a time. Existing works such as AudioCLIP (Guzhov et al., 2022) demonstrate the possibility of training an encoder for an additional modality such as audio on the side of the pre-trained frozen CLIP language-image encoders.

### 4.1 CONTRASTIVE LEARNING ON NEURAL NETWORK WEIGHTS

In our work, we consider multi-modal representations of meta-learning *tasks* $\mathcal{T}_i$, which may be presented in the form of language as task descriptions $t_i$, but potentially also in the form of images, videos, and audio. We fine-tune a base machine learning model $f(x, W_i) = y$ for task $\mathcal{T}_i$ and consider the base model as part of an alternative *meta-learning modality* for task $\mathcal{T}_i$. Fine-tuned networks from the *meta-learning modality* can then be paired in contrastive learning with the other multi-modal descriptions of $\mathcal{T}_i$. We can thus define our new **HyperCLIP** encoder as a "reverse hypernetwork" $CLIP_H(W_i) = e_i^{(H)}$, taking fine-tuned neural network weights $W_i$ as input, and outputting a CLIP embedding $e_i^{(H)}$ optimized for high cosine similarity with the CLIP embedding for the textual (and image, video, etc.) description of the task. We point to Figure 2 for an illustration of the approach, and to Algorithm 1 for the detailed training procedure.

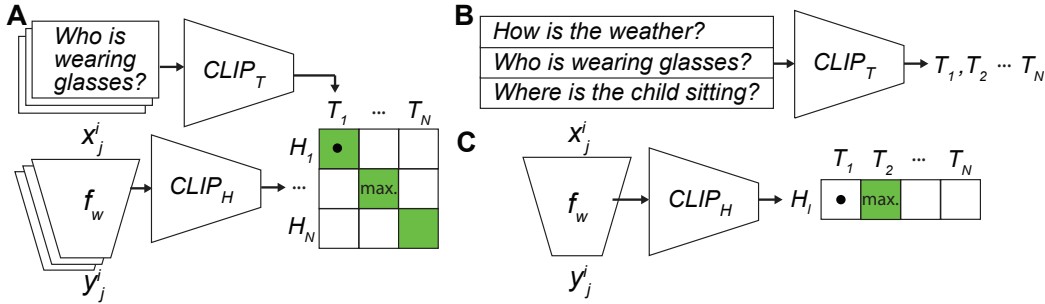

Figure 2: Our HyperCLIP encoder $\text{CLIP}_H$ is contrastively trained to map neural network weights $W$ to the latent space of a pre-trained language encoder $\text{CLIP}_T$, which we use to embed the natural language questions associated to the tasks (see **A**). To perform task inference given an already fine-tuned network, we encode all candidate task questions using the language CLIP encoder (see **B**), then encode the fine-tuned network weights with HyperCLIP (see **C**), and finally infer the correct task with a softmax operation over cosine similarities between HyperCLIP and language CLIP embeddings.

---

**Algorithm 1** HyperCLIP Training

---

sample a batch of tasks $\mathcal{T}_{i=1,\ldots,N}$ with loss functions $\mathcal{L}_{\mathcal{T}_i}$, training data $D_i^{\text{train}}$ and text $t_i$
define two $N$-sized arrays of $d$-dimensional embeddings $T \in \mathbb{R}^{N \times d}$ and $H \in \mathbb{R}^{N \times d}$
**for** $i = 1, \ldots, N$ **do**
    $T[i] = \text{CLIP}_T(t_i) \,/\, \|\text{CLIP}_T(t_i)\|$
    Fine-tune $W_i$ with objective: $\min_W \sum_{(x',y') \in D_i^{\text{train}}} \mathcal{L}_{\mathcal{T}_i}(f(x', W), y')$
    $H[i] = \text{CLIP}_H(W_i) \,/\, \|\text{CLIP}_H(W_i)\|$
**end for**
$\text{loss} = \left(\mathcal{L}_{\text{cross-entropy}}(TH^\top) + \mathcal{L}_{\text{cross-entropy}}(HT^\top)\right) / 2$
Update weights of $\text{CLIP}_H(.)$ using $\nabla \text{loss}$

---

## 4.2 CLASSIFIER-GUIDED META-LEARNING

On their own, CLIP encoders are at most useful for zero-shot or few-shot multi-modal classification as opposed to data generation. Recent popular image synthesis techniques, however, use CLIP encoders or other classifiers to *guide* generation from pre-trained unconditional generative models. *Classifier guidance* or *CLIP guidance* (Gal et al., 2021; Patashnik et al., 2021) use gradients with respect to a classifier or CLIP encoder to traverse a generative model's latent space.

In this work, we introduce **HyperCLIP guidance**, the first algorithm for classifier guidance in the meta-learning setting (Figure 1.B). Given a previously unseen validation task $\mathcal{T}_i$ and an unconditional generative hypernetwork model $h(z, \theta) = W$, we can use a trained HyperCLIP encoder $\text{CLIP}_H(W) = e^{(H)}$ to guide the exploration of the hypernetwork's latent space and find a set of base weights $W_i$ with high zero-shot performance for $\mathcal{T}_i$. Specifically, as long as we are given a starting hypernetwork latent vector $z^0$ and a textual description $t_i$ of the task, we can update $z^0$ with gradient descent over the guidance loss

$$\mathcal{L}_{\text{guidance}}(z) = -\frac{\text{CLIP}_H\left(h(z, \theta)\right)^\top \text{CLIP}_T(t_i)}{\|\text{CLIP}_H\left(h(z, \theta)\right)\| \|\text{CLIP}_T(t_i)\|} + \lambda \|z - z^0\|, \tag{4}$$

and then run the optimized latent vectors $\hat{z}_i$ through the generative hypernetwork to find adapted zero-shot base network weights $h(\hat{z}_i, \theta) = \hat{W}_i$ that perform well for the task.

## 5 HYPERLDM: TASK-CONDITIONAL DIFFUSION OF HYPERNETWORK LATENTS

Using classifier guidance to explore the latent space of a traditional generative model with direct CLIP gradients is a powerful and elegant approach. However, rapid innovation in the image synthesis

community recently led to this method being largely overcome in favor of applying classifier guidance and classifier-free guidance during the sampling process of a Diffusion Model (Dhariwal & Nichol, 2021; Ho & Salimans, 2021; Kim et al., 2022; Crowson, 2022; Nichol et al., 2022; Rombach et al., 2022). To paint a more complete picture, we also investigate this setting in the meta-learning domain, as an alternative to the previously introduced HyperCLIP guidance.

## 5.1 (LATENT) DIFFUSION MODELS

Denoising Diffusion Probabilistic Models (Sohl-Dickstein et al., 2015; Ho et al., 2020, DDPM) are a powerful class of generative models designed to learn a data distribution $p(x)$. They do so by learning the inverse of a *forward diffusion process* in which samples $x^0$ of our data distribution are slowly corrupted with additive Gaussian noise over $T$ steps with a variance schedule $\beta_1, \ldots, \beta_T$, resulting in the Markov Chain

$$q(x^t|x^{t-1}) = \mathcal{N}(x^t; \sqrt{1-\beta_t}x^{t-1}, \beta_t\mathbf{I}) \qquad q(x^{1:T}|x^0) = \prod_{t=1}^{T} q(x^t|x^{t-1}). \tag{5}$$

A property of such a process is that we can directly sample each intermediate step from $x^0$ as $x^t = \sqrt{\bar{\alpha}_t}x^0 + (\sqrt{1-\bar{\alpha}_t})\epsilon$ given $\epsilon \sim \mathcal{N}(0,\mathbf{I})$, $\alpha_t = 1 - \beta_t$ and $\bar{\alpha}_t = \prod_{s=1}^{t} \alpha_t$. Then, to learn the reverse process $p_\psi(x^{t-1}|x^t)$, we parametrize the timestep-dependent noise function $\epsilon_\psi(x^t, t)$ with a neural network and learn it by optimizing a simplified version of the variational lower bound on $p(x)$

$$\mathcal{L}_{\text{DM}}(\psi) = \mathbb{E}_{x,\epsilon\sim\mathcal{N}(0,1),t} \left[\|\epsilon - \epsilon_\psi(x^t, t)\|_2^2\right]. \tag{6}$$

Sampling from the reverse process can then be done with

$$x^{t-1} = \frac{1}{\sqrt{\alpha_t}} \left(x^t - \frac{\beta_t}{\sqrt{1-\bar{\alpha}_t}}\epsilon_\theta(x^t, t)\right) + \sigma_t\xi, \tag{7}$$

with $\xi \sim \mathcal{N}(0, \mathbf{I})$ and $\sigma_t$ chosen between $\beta_t$ and $\beta_t/\sqrt{1-\bar{\alpha}_t}$. Sampling from the learned diffusion model can be seen as analogue to Langevin Dynamics, a connection explicitly made in works exploring the diffusion technique from the "score matching" perspective (Song & Ermon, 2019; Song et al., 2020).

In our meta-learning setting, we aim to train a diffusion model which generates adapted zero-shot base network weights $\hat{W}_i$ that perform well for task $\mathcal{T}_i$. Thus, our diffusion model has to be conditional on a task embedding $e_i$. Moreover, in order to speed up training and leverage our previously trained generative hypernetwork $h(z, \psi)$, we define the diffusion process on latent vectors instead of doing so in weight space, emulating the Latent Diffusion technique from Rombach et al. (2022).

Satisfying the above requirements, we propose Hypernetwork Latent Diffusion Models (**HyperLDM**), which learn to sample from the conditional distribution of fine-tuned latent vectors $p(z^0|e_i)$ given a language CLIP embedding corresponding to the task. The HyperLDM neural network models the noise function $\epsilon_\psi(z^t, t, e_i)$, and is learned by optimizing the reweighted variational lower bound, which in this setting is

$$\mathcal{L}_{\text{LDM}}(\psi) = \mathbb{E}_{\mathcal{T}_i, h_{\text{enc}}(W_i), \epsilon\sim\mathcal{N}(0,1),t} \left[\|\epsilon - \epsilon_\psi(z^t, t, e_i)\|_2^2\right]. \tag{8}$$

## 5.2 CLASSIFIER-FREE GUIDANCE FOR META-LEARNING

The classifier guidance technique presented in Section 4.2 can be also adopted together with diffusion models. Even in the case of conditional diffusion models, the gradient of an auxiliary classifier (or CLIP encoder) can be added during sampling to induce an effect similar to GAN truncation (Brock et al., 2018), producing samples that are less diverse but of higher quality. Here, we showcase how the same technique can be adopted when generating adapted base network weights conditioned on task representations, tuning the temperature of sampling to reduce variance and increase base network performance.

The classifier-free guidance technique (Ho & Salimans, 2021; Nichol et al., 2022) allows us to leverage a conditional diffusion model to perform the same tempered sampling as above, without the auxiliary classifier. To do so, we train the conditional network $\epsilon_\psi(z^t, t, e_i)$ to also model the

unconditional case $\epsilon_\psi(z^t, t)$. One way of doing this is with *conditioning dropout*, simply dropping the conditional input $e_i$ for a certain percentage of training samples, inputting zeros instead. We can then sample at each diffusion iteration with

$$\tilde{\epsilon}_\psi(z^t, t, e_i) = (1 - \gamma)\,\epsilon_\psi(z^t, t, 0) + \gamma\epsilon_\psi(z^t, t, e_i). \tag{9}$$

For $\gamma = 0$, this recovers the unconditional diffusion model, while for $\gamma = 1$ it recovers the standard task-conditional model. For $\gamma > 1$, we instead obtain the classifier-free guidance effect, which we show results in the sampling of latent vectors $\hat{z}_i$ corresponding to higher-performing task-conditional network weights $h(\hat{z}_i, \psi) = \hat{W}_i$. We point to a more in-depth discussion on classifier-free guidance and its connection to score matching in Appendix A.1.

## 6 EXPERIMENTAL SETUP AND RESULTS

In this section, we demonstrate the soundness of our two approaches with zero-shot image classification experiments against a series of traditional meta-learning baseline techniques. Throughout our experiments, we fix the choice of base network model to a CLIP-Adapter model (see Appendix A.2), only varying the meta-learning techniques employed to obtain adapted base model weights.

### 6.1 THE META-VQA DATASET

To evaluate the performance of our methods, we utilize a dataset that reflects the setting of meta-learning with multi-modal task descriptors. Existing meta-learning benchmarks such as MiniImagenet (Ravi & Larochelle, 2016) or CIFAR-FS (Bertinetto et al., 2018) are unsuitable, as they are built for the traditional few-shot learning setting, in which the task $\mathcal{T}_i$ is not associated with task descriptors but is meant to be inferred through exposure to the support set $D_i^s$. We thus introduce our own **Meta-VQA** dataset, a modification of the VQA v2.0 dataset (Goyal et al., 2017) for Visual-Question-Answering. The dataset is composed of training and test tasks $\mathcal{T}_i$, each associated with a natural language question $t_i$ and a mini image classification dataset $(x_j^i, y_j^i) \in D_i$. We refer to Appendix A.3 for a more in-depth discussion.

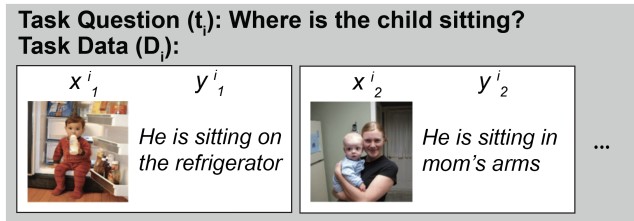

Figure 3: Example classification task from Meta-VQA, adapted from VQA v2 (Goyal et al., 2017). A single question $t_i$ is associated to multiple image-answer tuples $(x_j^i, y_j^i)$.

### 6.2 ZERO-SHOT TASK ADAPTATION WITH CLASSIFIER(-FREE) GUIDANCE

In Table 1 we show how our methods compare to a series of baselines when tested on the Meta-VQA dataset in the zero-shot setting. For each training task $\mathcal{T}_i$, the algorithms are given access to the full image/answer support and query sets $D_i^s, D_i^q$, together with the question (task descriptor) $t_i$. At test time, in the zero-shot setting, only the task descriptors $t_i$ for each test task $\mathcal{T}_i$ are given, and the algorithms are tasked with predicting the correct labels of images in the query set $D_i^q$. To simulate a setting in which we possess a larger "unconditional" pre-training dataset, we also train our model while only keeping a fraction of task descriptors from the Meta-VQA dataset: tasks without language descriptors can still be used to learn the unconditional HNET/HVAE model.

Classic zero-shot **CLIP** provides provide a 44.99% *floor* for performance on Meta-VQA, as our base model (CLIP-Adapter) consists of adapter layers over its output. We also can derive a ∼60.24% performance *ceiling* for zero-shot performance by considering the few-shot setting, in which models have also access to a data support set $D_i^s$ for every test task (see Appendix A.6): it is apparent that our zero-shot techniques cannot surpass this ceiling while keeping the choice of base model fixed.

Table 1: Zero-Shot learning accuracy averaged over Meta-VQA test tasks, with % of available question descriptors. Results should be interpreted as relative to a performance ceiling of ∼60.24 obtainable with few-shot learning on our fixed choice of base model (see Appendix A.6). (* ours)

| Method | Zero-shot (50% Q.) | Zero-Shot (100% Q.) |
|---|---|---|
| Base CLIP | 44.99 | |
| Uncond. Multitask | 53.75 (± 0.36) | |
| Uncond. MNet-MAML | 53.04 (± 0.69) | |
| Uncond. MNet-FOMAML | 53.04 (± 0.42) | |
| Uncond. HNet-MAML | 53.37 (± 0.29) | |
| Cond. Multitask | 51.68 (± 0.33) | 54.12 (± 0.80) |
| Cond. HNet-MAML | 51.54 (± 0.63) | 53.02 (± 0.20) |
| Cond. Multitask FiLM | 51.60 (± 0.56) | 53.84 (± 0.61) |
| * HNet + HyperCLIP Guidance | 53.51 (± 0.22) | 53.98 (± 0.54) |
| * HVAE + HyperCLIP Guidance | 53.82 (± 0.07) | 53.91 (± 0.08) |
| * HNet + HyperLDM $\gamma = 1$ | 53.66 (± 0.25) | 54.06 (± 0.21) |
| * HNet + HyperLDM $\gamma = 1.5$ | 54.08 (± 0.11) | 54.30 (± 0.27) |
| * HVAE + HyperLDM $\gamma = 1$ | 54.72 (± 0.23) | 55.03 (± 0.32) |
| * HVAE + HyperLDM $\gamma = 1.5$ | **54.84** (± 0.24) | **55.10** (± 0.08) |

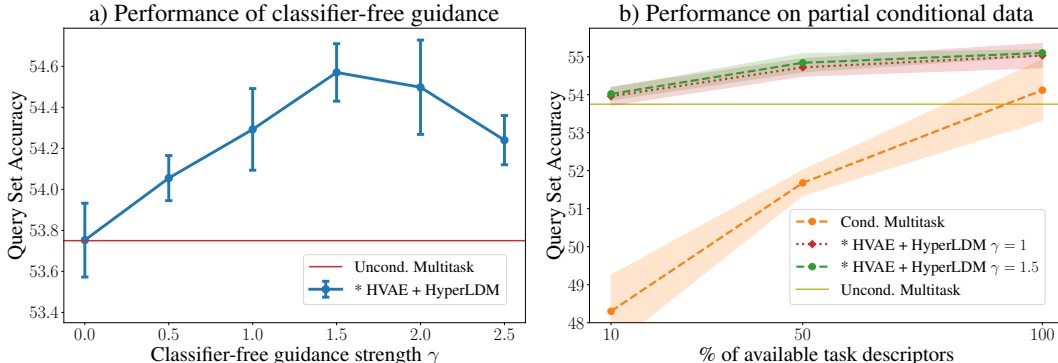

Figure 4: **a)** Performance of HyperLDM over different classifier-free guidance parameters $\gamma$. For $\gamma = 0$ we sample from an unconditional latent diffusion model. For $\gamma = 1$ we sample with classic conditioning. For $\gamma > 1$, we are in the classifier-free guidance regime. **b)** Performance of HyperLDM against baselines in the setting where only a fraction of natural language task labels are given.

We then benchmark several unconditional and conditional methods, with only conditional methods having access to language task descriptors. We apply MAML and its first-order variant FOMAML (Nichol et al., 2018) directly to the base network (**MNet-MAML**, **MNet-FOMAML**), and to both an unconditional hypernetwork (**Uncond. HNet-MAML**, as in Section 3) and a conditional one (**Cond. HNet-MAML**). We also benchmark against standard multitask learning (**Uncond. Multitask**, **Cond. Multitask**), and conditional multitask learning with the classic FiLM layer (Perez et al., 2017) (**Cond. Multitask FiLM**). It is apparent that the multitask approach, at least in this setting, leads to better zero-shot models than MAML, which instead optimizes for few-shot performance. We refer to Appendix A.2 and A.4 for more details on each model.

We then test **HyperCLIP Guidance** and **HyperLDM** when trained on top of either a hypernetwork or a VAE generator (see Appendix A.2 and A.5 for more detail). **HyperCLIP Guidance** allows for faster sampling than **HyperLDM** but is generally less performant, still, it performs on par with or slightly improves upon all other zero-shot baselines except for **Cond. Multitask**. The best performing model for the zero-shot setting is **HVAE + HyperLDM**, and specifically for classifier-free guidance with $\gamma = 1.5$. As illustrated in Figure 4.a, to further show the effectiveness of the classifier-free

guidance technique, we sweep over several candidate $\gamma$ parameters to find that the optimum occurs for $\gamma > 1$. As shown in Figure 4.b, when training our model while only keeping 50% or 10% of task descriptors, traditional **Cond. Multitask** learning is heavily impacted, while **HyperLDM** is almost not affected due to its two-phased training regime based on an uncondtional VAE. The gap between the multitask baseline and our HyperLDM technique is particularly striking in this setting.

## 7    RELATED WORK

**Hypernetworks**    By introducing multiplicative interactions within neural networks (Jayakumar et al., 2019), hypernetworks (Ha et al., 2016) have been shown to allow the modeling of diverse target network weights in, *e.g.*, continual learning, even in the compressive regime (von Oswald et al., 2021a; 2020) without loss of performance. For a given supervised problem, hypernetworks have been used to model the complex Bayesian posterior of the weights in conjunction with variational inference (Henning et al., 2018; Krueger et al., 2018). Similar approaches have been used for continual learning (Henning et al., 2021). Another vein of work consists in using hypernetworks to distill ensembles of diverse networks (Wang et al., 2018; Ratzlaff & Fuxin, 2020; von Oswald et al., 2021a).

**Meta learning**    In the context of meta-learning, hypernetworks have been successfully used in combination with popular gradient-based meta-learning methods (Finn et al., 2017; Zintgraf et al., 2019; Zhao et al., 2020; Flennerhag et al., 2020). More generally, related works have shown the usefulness of learning a low dimensional manifold in which to perform task-specific gradient-based adaptation at meta test time (Rusu et al., 2018; von Oswald et al., 2021b; Lee & Choi, 2018), instead of directly adapting in weight space. Recent works bypasses the formal bi-level formulation of meta-learning by, *e.g.*, using transformers to directly map the few-shot examples to the weights of the target network (Zhmoginov et al., 2022).

**Generative Modeling and Classifier(-free) guidance**    A plethora of techniques have been proposed for the generation of samples from high-dimensional domains such as images, such as Generative Adversarial Networks (Goodfellow et al., 2014; Brock et al., 2018, GANs) and Variational Autoencoders (Kingma & Welling, 2014, VAEs). Denoising Diffusion Probabilistic Models (Sohl-Dickstein et al., 2015; Ho et al., 2020, DDPM) overcome common issues in generative modeling using a simple likelihood-based reconstruction loss for iterative denoising, and have been shown to achieve state-of-the-art results in high resolution image generation (Dhariwal & Nichol, 2021; Rombach et al., 2022). Several techniques have been proposed for effective conditional sampling in generative and diffusion models, such as classifier/CLIP guidance (Dhariwal & Nichol, 2021; Gal et al., 2021; Patashnik et al., 2021) and classifier-free guidance (Ho & Salimans, 2021; Crowson, 2022; Nichol et al., 2022). Diffusion models with classifier-free guidance have also been successfully applied in non-visual domains, such as audio generation (Kim et al., 2022) and robotic planning (Janner et al., 2022).

**Zero-shot learning**    There exists a large literature on zero-shot learning, including both established benchmarks and well known methods (Han et al., 2021; Su et al., 2022; Gupta et al., 2021). While these zero-shot learning works consider the zero-shot performance on unseen class labels within a single classification task, our setting considers that of the zero-shot performance where test tasks themselves are unseen, thus raising the zero shot problem to the task-level.

## 8    CONCLUSION

In this work we introduced a framework that re-interprets meta-learning as a multi-modal generative modeling problem. Our HyperCLIP guidance and HyperLDM methods leverage this insight to generate task-adapted neural network weights in a zero-shot manner given natural language instructions, and constitute the first application of the CLIP guidance and classifier-free guidance techniques from image generation to the meta-learning domain. Our experiments show that our methods successfully make use of external task descriptors to produce high-performance adapted networks in the zero-shot setting.

ETHICS STATEMENT

Our contribution, despite showcasing experimental results, remains within the domain of general methods and thus does not face many immediate ethics concerns common in applied machine learning. Regardless, potential fairness and ethics issues should still be proactively investigated. A potential issue is that some of the methods described in the present work rely on pre-trained large models such as CLIP encoders, which may inadvertently encode harmful stereotypes and biases, in a manner which may not be immediately apparent to researchers and users making use of our downstream method.

REPRODUCIBILITY STATEMENT

We point to the supplementary material file attached to the submission for the complete codebase and Meta-VQA dataset tools, which can be used to reproduce our results. This code will be released online in more polished form after the requirement for anonymity is dropped. The code repository contains a `README.md` file with instructions for installing the necessary software environment. For a complete discussion of the hyperparameters used during our experimental runs, we refer to Appendix A.5. As for the Meta-VQA dataset, one must first download the original VQA v2 dataset (`https://visualqa.org/download.html`) and place it in the `data/VQA/` folder, and then run:

- `scripts/precompute_image_features.py`
- `scripts/precompute_ques_features.py`
- `scripts/precompute_text_features.py`

to re-generate the pre-computed CLIP embeddings.

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

# A APPENDIX

## A.1 CLASSIFIER-FREE GUIDANCE

We hereby provide a rationale for the use of classifier guidance and classifier-free guidance during diffusion model sampling. As per the "score matching" interpretation of diffusion models, we assume that our trained noise network approximates the score function of the true conditional latent distribution $p(z|e_i)$ as $\epsilon_\psi(z^t, t, e_i) \approx -\sigma_t \nabla_{z^t} \log p(z^t|e_i)$. For classifier guidance, we can perturb our diffusion sampling by adding the gradient of the log likelihood of our CLIP encoder $p_\psi(e_i|z_t)$ to the diffusion score as follows

$$\tilde{\epsilon}_\psi(z^t, t, e_i) = \epsilon_\psi(z^t, t, e_i) - \eta\sigma_t \nabla_{z^t} \log p_\psi(e_i|z^t) \approx -\sigma_t \nabla_{z^t} \left[ \log p(z^t|e_i) + \eta \log p_\psi(e_i|z^t) \right]. \tag{10}$$

We can rewrite this as classifier guidance on the unconditional score $\nabla_{z^t} \log p(z^t)$ with

$$-\sigma_t \nabla_{z^t} \left[ \log p(z^t) + \gamma \log p(e_i|z^t) \right] \quad \text{with} \quad \gamma = 1 + \eta \tag{11}$$

using Bayes' rule, as $\log p(z^t|e_i) = \log p(e_i|z^t) + \log p(z^t) - \log p(e_i)$, and thus $\nabla_{z^t} \log p(z^t|e_i) = \nabla_{z^t} \log p(e_i|z^t) + \nabla_{z^t} \log p(z^t)$.

For classifier-free guidance, we aim to perform the above sampling without access to a classifier, as long we possess a conditional diffusion model $\epsilon_\psi(z^t, t, e_i)$ that doubles as an unconditional model $\epsilon_\psi(z^t, t, 0)$, as illustrated in Section 5.2.

Using Bayes' rule again, we can see that $\nabla_{z^t} \log p(e_i|z^t) = \nabla_{z^t} \log p(z^t|e_i) - \nabla_{z^t} \log p(z^t)$. If we substitute this into Eq. 11 we obtain

$$-\sigma_t \nabla_{z^t} \left[ \log p(z^t) + \gamma \left( \log p(z^t|e_i) - \log p(z^t) \right) \right], \tag{12}$$

$$-\sigma_t \nabla_{z^t} \left[ (1-\gamma) \log p(z^t) + \gamma \log p(z^t|e_i) \right], \tag{13}$$

which can be implemented with our conditional network as

$$\tilde{\epsilon}_\psi(z^t, t, e_i) = (1-\gamma)\,\epsilon_\psi(z^t, t, 0) + \gamma\epsilon_\psi(z^t, t, e_i). \tag{14}$$

## A.2 NETWORK ARCHITECTURES

**Base Network ($f$)**   Our choice for a base model is a CLIP-Adapter (Gao et al., 2021), which consists of a frozen CLIP image encoder with added learned fully-connected layers refining the output embedding. Specifically, we use the *ViT-L/14@336px* CLIP encoder type with embedding size of 768. The advantages of this model choice lie in its combination of high base performance (due to pre-trained knowledge contained in the CLIP component) and relatively small parameter count, enabling agile medium-small scale experiments. This base CLIP-Adapter network purely works as a base model and is not to be confused with HyperCLIP, which is employed at the meta-level. In Section 6.2, when benchmarking the base model alone in the zero-shot setting, we drop the Adapter and use pre-trained zero-shot CLIP (Radford et al., 2021).

**Hypernetwork ($h$)**   For the hypernetworks used in our baseline as well as as the generative model, we use a MLP with one hidden layer of 256 units, which are followed by a rectified linear activation. For the unconditioned hypernetwork, the embedding to the hypernetwork is a vector of dimension 64, while for the conditioned counterpart, the task embedding is used. In order to ensure that the generated weights are properly normalized at initialization, we use the Kaiming initialization (He et al., 2015) for the hypernetwork weights, initialize the embedding as a sample from a multivariate standard gaussian distribution (for unconditioned models), and use the NTK parametrization (Jacot et al., 2020) for the target network.

**Variational Autoencoder**   For the variational autoencoder used as our unconditioned generative model, we use an MLP of 2 hidden layers of size 512 and 256, each followed by the rectified linear non-linearity. We chose 32 as the latent code dimension. We use the same architecture for the decoder, except for the dimensionality of the 2 hidden layers being swapped. We use the Kaiming initialization (He et al., 2015) to initialize the weight of both the encoder and decoder.

**HyperCLIP**   We parametrize our HyperCLIP model as a fully-connected MLP with a single hidden layer of dimension 256, taking as input the flattened weight of the base network and outputting the corresponding CLIP encoding. We chose the tangent hyperbolic function as the activation function in the hidden layer.

**HyperLDM**   While the original LDM makes use of a time-conditional UNet (Ronneberger et al., 2015) to parametrize the noise network, we are unfortunately unable to make use of spatial information and convolutions due to the non-spatial nature of our latent space. We parametrize our HyperLDM as a fully-connected network with residual connections and squeeze-and-excitation layers (Hu et al., 2018). The time index $t$ is embedded into a vector with a 150-dimensional sinusoidal positional embedding, and is concatenated together with the task-conditional embedding $e_i$ at the input layer and at intermediate activations. Hidden layer dimensions are 8192, 16384, 8192.

## A.3 THE META-VQA DATASET

The original VQA problem is about choosing a suitable natural language answer $a_k$ when prompted with both a natural language question $q_k$ and an image $I_k$. Our observation is that the VQA problem can then easily be reformulated as a meta-learning image classification problem with natural language task descriptions: given question-image-answer triples $(q_k, I_k, a_k) \in D$, we can group the data by unique questions $q_i$ (which will serve as task descriptor $t_i$), each of which can then be associated with supervised image classification tuples $(I_j^i, a_j^i) \in D_i$. To make sure the designed tasks are meaningful, we filter out question-answer pairs with questions in choosing form, *e.g.,* "A or B?" or yes/no answers. From the remaining questions we keep the ones which appear at least 20 times throughout the dataset, such that each task contains enough samples. In the end our Meta-VQA dataset is composed of 1234 unique tasks (questions), split into 870 training tasks and 373 test tasks, for a total of 104112 image-answer pairs. There are on average 9.13 answer choices per question/task. The average size of the support set is 57.85 examples, while the average size of the query set is 25.9 examples.

## A.4 BASELINE METHODS

We detail an overview of the baseline methods we benchmark in table 2, together with algorithm tables detailing each baseline method.

**Training:** The number of epochs each model is trained on, the learning rate `lr` of the optimization, as well as the learning rate and number of steps of the adaptation algorithm used for each method can be found in table 3. For all methods using an adaptation $\mathcal{A}_{\mathcal{T}_i}$, the dataset from the task is randomly split into a support set and a query set during training, every time a task is sampled. The support set is then used to perform the adaptation (see Section 2.1), while the query set is used to compute the loss on which the meta-parameters are updated. When no adaptation is used, all the data is used for this update. *Unconditional* methods do not have access to the task embedding $e_i$, while *conditioned* methods do. When the percentage of available task descriptor is reduced, conditioned methods are trained only on the tasks which descriptor is available.

**Evaluation:** Evaluation is performed on a fixed query set on the predefined query set of the held-out test tasks of the Meta-VQA dataset. Zero-shot performance is evaluated before applying the adaptation procedure $\mathcal{A}_{\mathcal{T}_i}$. For the few shot performance, all adaptation is performed on the support set of the test tasks. For MAML baselines, we keep the same adaptaiton-time learning rate as during training, while we always adapt for 50 steps. For each multitask baselines, we use the same adaptation scheme (steps, learning rate, adapting parameters) as their MAML counterpart.

---

**Algorithm 2** Unconditional Multitask Training

---

Define the base network $f$ with parameters $W$.
**for** epoch $= 1, \ldots, N$ **do**
    Sample a training batch of image-answer pairs $(x_k, y_k)$ from a mix of random training tasks $\mathcal{T}_i$.
    Update $W$ with gradient descent computed with respect to the classification loss over the sampled batch.
**end for**

---

**Algorithm 3** Unconditional MNet-MAML Training

---

Define the base network $f$ with parameters $W$.
**for** meta-epoch $= 1, \ldots, N$ **do**
    Sample a training task $\mathcal{T}_i$ and data $D_i$.
    Randomly split $D_i$ into support set $D_i^s$ and query set $D_i^q$.
    Run inner-loop adaptation $\mathcal{A}_{\mathcal{T}_i}$ using the support set $D_i^s$, fine-tuning $W$ into task-adapted $W_i = \mathcal{A}_{\mathcal{T}_i}(W)$.
    Use MAML gradient update to adapt $W$ given the inner-loop adaptation.
**end for**

---

**Algorithm 4** Unconditional HNet-MAML Training

---

Define the base network $f$ with parameters $W$.
Define a hypernetwork $h$ with meta-parameters $\theta$, mapping a latent vector $z^0$ to base network weights $W$.
**for** meta-epoch $= 1, \ldots, N$ **do**
    Sample a training task $\mathcal{T}_i$ and data $D_i$.
    Randomly split $D_i$ into support set $D_i^s$ and query set $D_i^q$.
    Run inner-loop adaptation $\mathcal{A}_{\mathcal{T}_i}$ using the support set $D_i^s$, fine-tuning $z^0$ into task-adapted $z_i = \mathcal{A}_{\mathcal{T}_i}(z^0)$.
    Use MAML gradient update to adapt $z^0$ and $\theta$ given the inner-loop adaptation.
**end for**

---

---

**Algorithm 5** Conditional Multitask Training

---

Define the base network $f$ with parameters $W$.

Define a hypernetwork $h$ with meta-parameters $\theta$, mapping the clip embedding $e_i$ of the language task descriptor to base network weights $W_i$.

**for** epoch $= 1, \ldots, N$ **do**

    Sample a training batch of task clip embedding, image and answer triples $(e_k, x_k, y_k)$ from a mix of random training tasks $\mathcal{T}_i$.

    Update $\theta$ with gradient descent computed with respect to the classification loss over the sampled batch.

**end for**

---

**Algorithm 6** Conditional HNet-MAML Training

---

Define the base network $f$ with parameters $W$.

Define a hypernetwork $h$ with meta-parameters $\theta$, mapping the clip embedding $e_i$ of the language task descriptor to base network weights $W_i$.

**for** meta-epoch $= 1, \ldots, N$ **do**

    Sample a training task $\mathcal{T}_i$, data $D_i$ and the clip embedding $e_i$ of the task descriptor.

    Randomly split $D_i$ into support set $D_i^s$ and query set $D_i^q$.

    Run inner-loop adaptation $\mathcal{A}_{\mathcal{T}_i}$ using the support set $D_i^s$, fine-tuning $e_i$ into task-adapted $\tilde{e}_i = \mathcal{A}_{\mathcal{T}_i}(e_i)$.

    Use MAML gradient update to adapt $\theta$ given the inner-loop adaptation.

**end for**

---

Table 2: Overview of the different methods trained on MetaVQA. The **parameters** are optimized via the task loss evaluated on the output of the **function**, averaged over minibatches of tasks. The adaptation $\mathcal{A}_{\mathcal{T}_i}$ implements a few step gradient descent algorithm applied on the argument parameter, w.r.t the task loss evaluated on the support set.

| Method | Function | Parameters |
|---|---|---|
| Unconditional Multitask | $f(\cdot, W)$ | $W$ |
| Unconditional MNet (FO)MAML | $f(\cdot, \mathcal{A}_{\mathcal{T}_i}(W^0))$ | $W^0$ |
| Unconditional Hypernetwork MAML | $f(\cdot, h(\mathcal{A}_{\mathcal{T}_i}(z^0), \theta))$ | $\theta, z^0$ |
| Conditional Multitask | $f(\cdot, h(e_i, \theta))$ | $\theta$ |
| Conditional Hypernetwork MAML | $f(\cdot, h(\mathcal{A}_{\mathcal{T}_i}(e_i), \theta))$ | $\theta$ |

Table 3: Hyperparameters used for the baseline methods. All methods are trained with the Adam (Kingma & Ba, 2017) optimizer, with meta-batch size of 32 tasks. We use gradient norm clipping for all optimization, with the maximum norm set to 10. Note that when the adaptation algorithm has a range of possible steps, the number of step is sampled uniformly from the range for every adaptation.

| Method | epochs | lr | $\mathcal{A}$-lr | $\mathcal{A}$-steps |
|---|---|---|---|---|
| Unconditional MNet Multitask | 300 | 0.0001 | - | - |
| Unconditional MNet (FO)MAML | 500 | 0.00003 | 0.01 | 0-10 |
| Unconditional Hypernetwork MAML | 100 | 0.00003 | 0.1 | 0-10 |
| Conditional Multitask | 60 | 0.0001 | - | - |
| Conditional Hypernetwork MAML | 200 | 0.00001 | 0.1 | 0-10 |

## A.5 GUIDANCE MODELS

### A.5.1 GENERATIVE HYPERNETWORK

To enable our guidance methods, we need to first train a generative hypernetwork $h$ as in Section 3, either in the form of an Unconditional Hypernetwork, or of a Hypernetwork VAE:

- For **HNet + HyperCLIP guidance** and **HNet + HyperLDM**, we meta-learnt an unconditioned hypernetwork with the exact same hyperparameters as the baseline **Uncond. HNet-MAML**, and used it as the generative hypernetwork.

- For **HVAE + HyperCLIP guidance** and **HVAE + HyperLDM**, we trained an unconditioned VAE on samples of fine tuned network weights $W_i$ using the architecture specified in A.2. In order to be able to quickly sample new adapted weights, and to reduce the complexity of the manifold from which such weights are sampled, we use adaptations from our unconditional MAML baselines as $W_i$. Specifically, *Variant 1* (Algorithm 7 involves adaptations from **Uncond. HNet-MAML**, using 50-step adaptation $\mathcal{A}_{\mathcal{T}_i}$ with learning rate 0.1, on support set stochastically sampled for every adaptation phase. *Variant 2* (Algorithm 8) involves adaptations over the base network (initialized from a learned **Uncond. MNet-MAML** initialization), using 50-step adaptation $\mathcal{A}_{\mathcal{T}_i}$ with learning rate 0.01, on support set stochastically sampled for every adaptation phase. We trained the VAE on 2000 epochs where each epoch is a single pass through all the tasks, with the Adam (Kingma & Ba, 2017) optimizer and 0.0001 learning rate and batch size 32. We used gradient norm clipping independently for both the encoder and decoder, with the maximum norm capped at 1000. For the final results presented in our main results table (Table 1), we use Variant 2.

---

**Algorithm 7** HVAE Training, Variant 1

---

Define the base network $f$ with parameters $W$.
Define an encoder $z = d(W, \omega)$ with parameters $\omega$ and a hypernetwork decoder $W = h(z, \theta)$ with parameters $\theta$.
Obtain a previously learned HNet $\tilde{h}(z^0, \tilde{\theta})$ according to **Uncond. HNet-MAML** (Algorithm 6).
**for** epoch $= 1, \ldots, N$ **do**
    Create an empty batch $B = \{\}$.
    **for** $b = 1, \ldots, M$ **do**
        Sample a training task $\mathcal{T}_i$ and data $D_i$.
        Randomly split $D_i$ into support set $D_i^s$ and query set $D_i^q$.
        Run inner-loop adaptation $\mathcal{A}_{\mathcal{T}_i}$ using the support set $D_i^s$, fine-tuning $z^0$ from the HNet $\tilde{h}$ into task-adapted $\tilde{z}_i = \mathcal{A}_{\mathcal{T}_i}(z^0)$.
        Given fine-tuned $z_i$, produce task-adapted weights $W_i = \tilde{h}(z_i, \tilde{\theta})$.
        Add the fine-tuned weights to the batch: $B = B \cup \{W_i\}$.
    **end for**
    Train the HVAE encoder and decoder using the VAE loss to reconstruct the weight batch $B$.
**end for**

---

---

**Algorithm 8** HVAE Training, Variant 2

---

Define the base network $f$ with parameters $W$.
Define an encoder $z = d(W, \omega)$ with parameters $\omega$ and a hypernetwork decoder $W = h(z, \theta)$ with parameters $\theta$.
Obtain a previously learned base network initialization $W^0$ according to **Uncond. MNet-MAML** (Algorithm 6).
**for** epoch $= 1, \ldots, N$ **do**
    Create an empty batch $B = \{\}$.
    **for** $b = 1, \ldots, M$ **do**
        Sample a training task $\mathcal{T}_i$ and data $D_i$.
        Randomly split $D_i$ into support set $D_i^s$ and query set $D_i^q$.
        Run inner-loop adaptation $\mathcal{A}_{\mathcal{T}_i}$ using the support set $D_i^s$, fine-tuning $W_i = \mathcal{A}_{\mathcal{T}_i}(W^0)$.
        Add the fine-tuned weights to the batch: $B = B \cup \{W_i\}$.
    **end for**
    Train the HVAE encoder and decoder using the VAE loss to reconstruct the weight batch $B$.
**end for**

---

### A.5.2 HYPERCLIP

**Training** In order to train the HyperCLIP model, we need samples of fine tuned network weights $W_i$. Similarly to HVAE, we used adaptations from **Uncond. HNet-MAML**, using 50-step adaptation $\mathcal{A}_{\mathcal{T}_i}$ with learning rate 0.1, on a support set stochastically sampled at every adaptation phase, as this would allow us to use the same HyperCLIP model for doing guidance on both HNet and HVAE. We trained our HyperCLIP model for 600 epochs, with the Adam (Kingma & Ba, 2017) optimizer, 0.0003 learning rate and batch size 64, for all our experiments.

**Guidance** We use 10 steps guidance with $\lambda = 0.01$ and learning rate 0.1, for both when performed on HNet and HVAE.

**Evaluation** Evaluation is performed on a fixed query set on the predefined query set of the held-out test tasks of the Meta-VQA dataset. Zero-shot performance is evaluated on the output of the generative hypernetwork $h$ after applying latent space guidance. For the few-shot performance, all adaptation is performed on the support set of the test tasks, on the latent space initialized at the output of the guidance procedure. Similarly to our baselines, we use 50-steps gradient descent adaptation with learning rate 0.1.

---

**Algorithm 9** HNet + HyperCLIP Training

---

Learn an unconditional hypernetwork $h(z^0, \theta)$ with the **Uncond. HNet-MAML** procedure from Algorithm 6.
Learn HyperCLIP network $\text{CLIP}_H(W)$ using the HyperCLIP training procedure from Algorithm 1. For sampling fine-tuned $W_i$, fine-tune the base-network on training tasks.

---

**Algorithm 10** HVAE + HyperCLIP Training

---

Learn an unconditional hypernetwork $h(z, \theta)$, as the decoder of a HVAE (Algorithm 8).
Learn HyperCLIP network $\text{CLIP}_H(W)$ using the HyperCLIP training procedure from Algorithm 1. For sampling fine-tuned $W_i$, fine-tune the base-network on training tasks.

---

---

**Algorithm 11** HyperCLIP Guidance (Inference time)

---

Define a learned unconditional hypernetwork $h(z, \theta)$, as either a HNet $h(z^0, \theta)$ (Algorithm 6) or the decoder of a HVAE (Algorithm 8).
Define a learned HyperCLIP network $\text{CLIP}_H(W)$.
Define an unseen task $\mathcal{T}_i$ with natural language task descriptor $t_i$.
Randomly sample $z \sim \mathcal{N}(0, I)$ if using the decoder of a HVAE, or set $z = z^0$ where $z^0$ is the meta learned embedding initialization of the Hnet.
Optimize $z$ with gradient descent over $\mathcal{L}_{\text{guidance}}(z)$ (Eq. 4), obtaining *guided* $z_i$.
Obtain *guided* base weights $W_i = h(z_i, \theta)$.
Use adapted base network $f$ with weights $W_i$ to classify examples from the unseen task $\mathcal{T}_i$.

---

### A.5.3 HYPERLDM

**Training**  Similarly to HyperCLIP, to train HyperLDM we need samples of fine tuned network weights $W_i$, for which we use adaptations from **Uncond. HNet-MAML**, using 50-step adaptation $\mathcal{A}_{\mathcal{T}_i}$ with learning rate 0.1, on a support set stochastically sampled at every adaptation phase. We parametrize the diffusion process with a linear noise schedule, $\beta$ starting at 0.0001 and ending at 0.06, and 350 diffusion timesteps. We train the HyperLDM for 1000 epochs with the Adam optimizer, 0.00025 learning rate and 128 epochs, for all our experiments.

**Evaluation**  Evaluation is performed as for HyperCLIP guidance, except for the fact that adaptation is performed natively through sampling from the learned reversed diffusion process, with parameters derived from the chosen $\beta$ schedule. The guidance parameter $\gamma > 0$ can be tuned during inference to accentuate the effect of classifier-free guidance.

---

**Algorithm 12** HNet + HyperLDM Training

---

Learn an unconditional hypernetwork $h(z^0, \theta)$ with the **Uncond. HNet-MAML** procedure from Algorithm 6.
Learn the HyperLDM network $\epsilon_\psi(z^t, t, e_i)$ using the HyperLDM training procedure, optimizing reconstruction of $z_i^0$ with loss from Eq. 8. For sampling fine-tuned $z_i$, fine-tune the base-network on training tasks, then encode the weights using the HNet.

---

**Algorithm 13** HVAE + HyperLDM Training

---

Learn an unconditional hypernetwork $h(z, \theta)$, as the decoder of a HVAE (Algorithm 8).
Learn the HyperLDM network $\epsilon_\psi(z^t, t, e_i)$ using the HyperLDM training procedure, optimizing reconstruction of $z_i^0$ with loss from Eq. 8. For sampling fine-tuned $z_i$, fine-tune the base-network on training tasks, then encode the weights using the HVAE.

---

**Algorithm 14** HyperLDM Inference

---

Define a learned unconditional hypernetwork $h(z, \theta)$, as either a HNet $h(z^0, \theta)$ (Algorithm 6) or the decoder of a HVAE (Algorithm 8).
Define a learned HyperLDM network $\epsilon_\psi(z^t, t, e_i)$.
Define an unseen task $\mathcal{T}_i$ with natural language task descriptor $t_i$, with clip embedding $e_i$.
Randomly sample $z \sim \mathcal{N}(0, I)$.
Iteratively modify $z$ with diffusion sampling using the learned $\epsilon_\psi$ network, obtaining *guided* $z_i$.
Obtain *guided* base weights $W_i = h(z_i, \theta)$.
Use adapted base network $f$ with weights $W_i$ to classify examples from the unseen task $\mathcal{T}_i$.

---

### A.6 FEW-SHOT LEARNING

For completeness, we include in Table 4 the results for few-shot learning on the test split of Meta-VQA. Our technique, unlike classic MAML, does not optimize specifically for the few-shot learning

setting. Instead, the few-shot learning results are meant to contextualize performance gains in the zero-shot setting: zero-shot performance gains should be interpreted as relative to the few-shot performance ceiling of 60.24, the maximum attained with our fixed choice of base model.

Table 4: Few-Shot learning accuracy averaged over Meta-VQA test tasks. (* ours)

| Method | Few-Shot |
|---|---|
| Base CLIP-Adapter | 54.93 ($\pm$ 0.11) |
| Uncond. Multitask | 55.53 ($\pm$ 0.40) |
| Uncond. MNet-MAML | **60.24** ($\pm$ 0.84) |
| Uncond. MNet-FOMAML | 60.03 ($\pm$ 0.48) |
| Uncond. HNet-MAML | 58.70 ($\pm$ 0.10) |
| Cond. Multitask | 59.46 ($\pm$ 0.31) |
| Cond. HNet-MAML | 59.48 ($\pm$ 0.03) |
| * HNet + HyperCLIP Guidance | 58.82 ($\pm$ 0.27) |
| * HVAE + HyperCLIP Guidance | 58.75 ($\pm$ 0.29) |
| * HNet + HyperLDM $\gamma = 1$ | 58.70 ($\pm$ 0.11) |
| * HNet + HyperLDM $\gamma = 1.5$ | 58.60 ($\pm$ 0.09) |
| * HVAE + HyperLDM $\gamma = 1$ | 58.97 ($\pm$ 0.09) |
| * HVAE + HyperLDM $\gamma = 1.5$ | 58.89 ($\pm$ 0.07) |

