# OpenReview forum: "Meta-Learning via Classifier(-free) Guidance"
_ICLR.cc/2023/Conference — Submitted to ICLR 2023_

### Official Review · Reviewer_5YoJ · 2022-10-23

**Confidence:** 4
**Correctness:** 3
**Technical Novelty And Significance:** 3
**Empirical Novelty And Significance:** 1
**Recommendation:** 3

**Clarity, Quality, Novelty And Reproducibility:**

- Reproducibility: The submission came with code in the supplementary, a plus for reproducibility.
- Novelty: The model is novel but empirical evidence suggests that it is not very useful.
- Clarity: The paper writing is ok, but Figure 1 is very intimidating. It also lacks a central hypothesis of what it is trying to show. Why a hyper clip guidance with latent diffusion in the first place?

**Strength And Weaknesses:**

Strengths:
- The authors made a huge effort designing many model components and have spent time writing a paper that describes their idea.
- The authors also contributed a new evaluation dataset based on existing data.

Weaknesses:
- Complexity vs. significance. Despite a plethora of new components proposed, e.g. VAE, HyperClip, Guidance, Hyper latent diffusion etc, the method does better than baselines only by 0.5% in zero-shot meta-VQA, which is surprisingly marginal. I suggest that the authors find a minimal set of modifications with the maximal difference of results.
- Although it is good to see a more realistic dataset on VQA proposed, the paper could also be evaluated on traditional zero-shot and few-shot tasks.
- The paper lacks a thorough description of how the meta-VQA dataset is created. And also there is no visualization of the examples, nor is there qualitative analyses. I also checked Appendix A.3 and it is still very unclear. A few more examples with model predicted outputs/groundtruth will be very helpful to understand the setup.

**Summary Of The Paper:**

This paper introduces a hypernetwork that generates the adapted weights for each task and runs the experiment on a adapted dataset called “meta VQA” for testing zero-shot and few-shot performance. The results suggest that the added benefit of the proposed module is tiny.

**Summary Of The Review:**

In summary, the experimental results show that it is probably unnecessary to introduce so many model complications. The newly introduced dataset is a highlight, but the paper could have been evaluated on existing few-shot / zero-shot datasets too. Therefore, I think it is definitely below the bar of acceptance.

---
I acknowledge that I have read the rebuttal. I thank the authors for providing additional results and raise my score from 1 to 3. Overall, the paper lacks a great amount of clarity in terms of model design and dataset evaluation, and the model design seems intimidating while the added benefits are small.

---

> ### Author Response · Authors · 2022-11-11
> **Response to Reviewer 5YoJ**
>
> We thank the reviewer for the careful evaluation of our work, and providing overall positive feedback on its correctness and technical novelty/significance. We will address the remaining concerns one by one and believe that the additional clarification can convince the reviewer to vote for an acceptance of the paper.
>
> **P1**:
> > Despite a plethora of new components proposed, e.g. VAE, HyperClip, Guidance, Hyper latent diffusion etc, the method does better than baselines only by 0.5% in zero-shot meta-VQA, which is surprisingly marginal. I suggest that the authors find a minimal set of modifications with the maximal difference of results.
>
> **A1**: We point to our general response for the clarification on the strength of our method, as well as our new experimental results. In the experiment where a certain percentage of task descriptor is removed, our method achieves 54.84% **zero-shot accuracy** on unseen tasks, whereas the multitask baselines only achieve 51.68%. This improvement in performance is all the more significant when taking into account that MAML only achieves 60.24% accuracy on the same set, **even after using few-shot examples** (which our method does not use). We hope the reviewer can agree that this result alone is already empirically significant.
>
> **P2**:
> > Although it is good to see a more realistic dataset on VQA proposed, the paper could also be evaluated on traditional zero-shot and few-shot tasks
>
> **A2**: Thank you for raising this point. We agree that evaluating the performance on other benchmarks would have been desirable. However, the setting we consider in our work is, to the best of our knowledge, unconventional and lacks a suitable benchmark, which motivated the construction of the VQA dataset. While traditional zero-shot learning settings consider the zero-shot performance on e.g. unseen class labels within a single classification task, we consider in our work the zero-shot performance on unseen tasks (i.e. where the question being asked itself changes, as opposed to simple “object detection”). We do not think there are obvious ways to apply existing zero-shot methods to this setting, nor do we think we can repurpose existing zero-shot benchmarks for our method.
>
> **P3**:
> > The paper lacks a thorough description of how the meta-VQA dataset is created. And also there is no visualization of the examples, nor is there qualitative analyses. I also checked Appendix A.3 and it is still very unclear. A few more examples with model predicted outputs/groundtruth will be very helpful to understand the setup.
>
> **A3**: We apologize if our description of the Meta-VQA dataset in Appendix A.3 was not exhaustive. We have improved it with additional context. As mentioned in Section 6.1, the Meta-VQA dataset is created as a simple modification of the VQA v2.0 dataset (Goyal et a., 2017), performing a “grouping” operation over identical questions, such that each question constitutes a “task”, with its associated support and query sets, each constituted of image-answer pairs (x,y). It is unclear to us what is meant by “no visualization of the examples”, as Figure 3 exactly illustrates such a task example, with image-answer example pairs (a picture of a held toddler should be classified as “He is sitting in mom’s arms”, etc.).
> We added some key dataset quantities to Appendix A.3. As for “qualitative analyses”, it is unclear to us what is meant exactly.
>
> **P4**:
> > The paper writing is ok, but Figure 1 is very intimidating. It also lacks a central hypothesis of what it is trying to show. Why a hyper clip guidance with latent diffusion in the first place?
>
> **A4**: Figure 1 is meant to illustrate our novel learning architecture contributions. As for why we choose our approach, we believe to answer that thoroughly in the introduction of our paper: we believe that recent advances in the field of generative modeling (and specifically image synthesis) can help us in other settings and “modalities”. Specifically, the separation of a “pre-training” phase and a “guidance” phase common in language-conditioned image synthesis can be repurposed for zero-shot meta-learning. Applying these techniques allows us to design interesting algorithms, while resulting in higher performance on our newly introduced Meta-VQA task compared to state of the art multitask learning methods.
>
>
> We thank the reviewer again for the constructive feedback, which made us realize the lack of clarity around our central empirical result. We hope that given the additional clarification, together with the renewed strong empirical evidence provided in the general response, the reviewer will reconsider the low empirical novelty/significance score and consider voting for acceptance of the paper. If there are any points that need further clarification or improvement, we would of course be happy to address them during the remainder of the rebuttal period.

---

> ### Author Response · Authors · 2022-11-18
> **Reminder**
>
> We once again thank you for your review and suggestions. We apologize for the insistence, but we would like to gently remind you that the interactive period will expire in less than a day. We would like to engage in a discussion about any aspects of the paper or of our rebuttal that you may still object to. For this reason, we would like to have the opportunity to read your further replies before our ability to respond is blocked.
> Thanks a lot again, and we apologize for the inconvenience.

---

### Official Review · Reviewer_mvke · 2022-10-24

**Confidence:** 4
**Clarity, Quality, Novelty And Reproducibility:** See above
**Correctness:** 2
**Technical Novelty And Significance:** 2
**Empirical Novelty And Significance:** 2
**Recommendation:** 3

**Strength And Weaknesses:**

While the use of hyper-networks in meta learning is not new, this particular choice of architecture and loss (VAE+ELBO) in this paper has not been investigated before. The proposed methods for zero-shot learning, in particular the LDM based variant, is novel to my best knowledge.

I have some concerns regarding the design of the method and the experiments, listed below:

1. The VAE has the critical role of compressing and linearizing network weights into an euclidean-like latent space. Have you investigated the choice of architecture for both the encoder and decoder beyond shallow MLPs? Often we see adaptive-normalization layers being more appropriate than feedforward MLPs for outputting to non-euclidean vector spaces, have you experimented with any such models?

2. There exists a large existing literature on zero-shot learning, including both established benchmarks (e.g. OpenImages, Cub200-2011, Animals with Attributes AWA, imagenet 21k etc.) and well known methods (e.g. https://arxiv.org/abs/2101.11606v2, https://openaccess.thecvf.com/content/CVPR2022/html/Su_Distinguishing_Unseen_From_Seen_for_Generalized_Zero-Shot_Learning_CVPR_2022_paper.html, https://openaccess.thecvf.com/content/CVPR2022/html/Zhu_Dual_Cross-Attention_Learning_for_Fine-Grained_Visual_Categorization_and_Object_Re-Identification_CVPR_2022_paper.html, https://openaccess.thecvf.com/content/CVPR2021/html/Han_Contrastive_Embedding_for_Generalized_Zero-Shot_Learning_CVPR_2021_paper.html). Have you 1. experimented with running either of your proposed methods on standard ZSL benchmarks? And 2. Have you tried running any previous works (besides CLIP) on your proposed dataset? The related works section also do not cite any prior works in zero-shot learning.

3. Very few details are provided regarding the definition of the baselines that you compare to. For example, how is MAML used without any test-time adaptation? How do you solve the class assignment problem in that case? What is the “standard multitask learning method” referring to? How are the multitask methods adapted for the few-shot setting?

4. Characteristics of the proposed meta-VQA datasets are missing: how many choices are there on average for each question? How do you split the training and test tasks (e.g. did you check if semantically identical questions exist between training and testing)? How big is the support set in the few-shot setting?

5. It is concerning that the performance of the proposed methods do not significantly out-perform baselines on the only zero-shot dataset evaluated, and is even out-performed on the few-shot task. I would suspect that if baselines designed for ZSL are included, the comparisons would appear even worse. Can you provide any hypothesis as to under what circumstances will the proposed method exhibit quantifiably better performance than existing methods?

6. Diffusion models, particularly when imbued with sufficient capacity, are known to overfit rapidly to small datasets, where latents are all mapped to known training examples. Have you validated the generative capabilities of your diffusion model? Is it really working as intended?


**Summary Of The Paper:**

This work presents two new approaches to zero-shot learning via building generative models of classifier weights. Contrastive learning is used to build a CLIP-style model which matches task description with network weights. In one approach, this CLIP-style model is used at test time to maximize match between provided task description and the network weight. Here, the search space for network weights is constrained to the latent space of the generative model.
In the other approach, a DM is trained on the latent space of the generative model, conditioned on the CLIP embedding of the task. Classifier-free guidance is used at sampling time to improve quality. A new dataset Meta-VQA is also proposed for evaluating zero-shot classification methods.

**Summary Of The Review:**

Given the above concerns, I think this work is not suitable for publication in its current state, and hence I suggest rejection.

---

> ### Author Response · Authors · 2022-11-11
> **Response to Reviewer mvke (1/2)**
>
> We thank the reviewer for carefully evaluating our work. We have particularly appreciated the accurate summary of our paper. We address your concerns point by point below.
>
> **P1**:
> > Have you investigated the choice of architecture for both the encoder and decoder beyond shallow MLPs? Often we see adaptive-normalization layers being more appropriate than feedforward MLPs for outputting to non-euclidean vector spaces, have you experimented with any such models?
>
> **A1**: Thank you for pointing this out. We have indeed not heavily investigated architectural choices for the VAE other than a simple feed forward one. We believe adding additional inductive biases, such as symmetries of neural networks weights, or adaptive normalization layers, would help the VAE better model the weight distribution, and ultimately help improve the end result. Indeed, defining good inductive biases for hypernetworks is an open problem that is very much of interest to us, and we are not aware of existing work that satisfyingly solves it. We are currently training the VAE with adaptive normalization layers as you suggested, and will use the result in the final version of our paper if the performance improves.
>
> **P2**:
> > Have you 1. experimented with running either of your proposed methods on standard ZSL benchmarks? And 2. Have you tried running any previous works (besides CLIP) on your proposed dataset? The related works section also do not cite any prior works in zero-shot learning.
>
> **A2**: Thank you for raising these points. Unless we misunderstood your comment, we do not see a straightforward way to apply these existing zero-shot learning methods to our setting. ZSL in general considers the zero-shot performance on e.g. unseen class labels within a single (potentially multilabel) classification task. In our work, we consider the zero-shot performance where test tasks themselves are unseen, raising the zero shot problem to the task-level (i.e. the “question” we ask during classification changes, such as “who is wearing glasses?”, “where is the child sitting?”, problems that involve more than object detection to be solved). Instead of leveraging class descriptors to classify new classes using trained features, our setting consists in leveraging task descriptors to change the features for the classification problem of new tasks. It is not obvious to us how to apply existing zero-shot methods to this setting, nor do we think we can repurpose existing zero-shot benchmarks for our method. We remain of course very interested to hear any suggestions or clarification that you may have.
> We nevertheless added a ZSL paragraph to our related work section, and explicitly contrasted our setting to improve clarity.
>
> **P3**:
> > For example, how is MAML used without any test-time adaptation?
>
> **A3**: For the MAML baselines, the MAML algorithm is applied at training time to learn a network initialization. Zero-shot performance is then evaluated on this initialization before applying the adaptation procedure. We realize that MAML is not optimizing for zero-shot performance, but we decided to include its zero-shot performance for completeness, given that MAML is what we use to obtain the few-shot performance ceiling of ~60% for our choice of base model.
>
> **P4**:
> > How do you solve the class assignment problem in that case?
>
> **A4**: Our choice of base model is detailed in Appendix A.2, in the “Base Network” chapter. This CLIP-Adapter model, as described in the cited literature, performs classification with a softmax operation over the vector of cosine distances between the image’s CLIP-adapted embedding and the language labels’ CLIP embeddings.
> Our method produces neural network weights for the Adapter component of the CLIP-Adapter base model.
>
> **P5**:
> > What is the “standard multitask learning method” referring to? How are the multitask methods adapted for the few-shot setting?
>
> **A5**: We recognize that we did not describe the multitask baselines in enough detail, for this reason, we updated Appendix A.4 with a more in-depth description of them. The “standard multitask learning method” consists of fine-tuning the base network directly on all training tasks, with training batches comprised of shuffled samples from any training task, without an outer-loop meta-learning process. In the Unconditional multitask learning baseline, the base network is simply fine-tuned on image-answer data from all tasks, while the Conditional multitask learning baseline allows the base network to take in as an additional input the language embedding obtained from the language task descriptor (question).
> For each multitask baseline we use the same adaptation scheme (steps, learning rate, adapting parameters) as we do for the inner-loop in their MAML counterpart.

---

> > ### Author Response · Authors · 2022-11-11
> > **Response to Reviewer mvke (2/2)**
> >
> > **P6**:
> > > Characteristics of the proposed meta-VQA datasets are missing: how many choices are there on average for each question? How do you split the training and test tasks (e.g. did you check if semantically identical questions exist between training and testing)? How big is the support set in the few-shot setting?
> >
> > **A6**: We thank you for raising this point, we realize that we may not have been exhaustive enough in describing the details of the Meta-VQA dataset. Over the entire dataset, there are on average 9.13 choices per question/task. Training and test tasks have been randomly split, with 870 tasks in the training set and 373 in the test set. As the questions come from the VQA v2 dataset (Goyal et al., 2019), they are formulated following a common syntactic formula (for easy indexing) which allows there not to be separate questions which are semantically identical. The average size of the support set is 57.85 examples, while the average size of the query set is 25.9 examples.
> >
> > **P7**:
> > > It is concerning that the performance of the proposed methods do not significantly out-perform baselines on the only zero-shot dataset evaluated, and is even out-performed on the few-shot task. I would suspect that if baselines designed for ZSL are included, the comparisons would appear even worse. Can you provide any hypothesis as to under what circumstances will the proposed method exhibit quantifiably better performance than existing methods?
> >
> > **A7**: We point to our general response for the clarification on the strength of our method, as well as our new experimental results. In particular, in our experiment where a certain percentage of task descriptor is removed, our method achieves 54.84% **zero-shot accuracy** on unseen tasks, whereas the multitask baselines only achieve 51.68%. This improvement in performance is all the more significant when taking into account that MAML only achieves 60.24% accuracy on the same set **after using few-shot examples** (which our method does not use). We hope that our changes and clarifications answer your concerns about a lack of clear advantages compared to baselines. As for your comment regarding ZSL baselines, we believe we cannot compare ZSL to our method for the reason specified in answer A2.
> >
> > **P8**:
> > > Diffusion models, particularly when imbued with sufficient capacity, are known to overfit rapidly to small datasets, where latents are all mapped to known training examples. Have you validated the generative capabilities of your diffusion model? Is it really working as intended?
> >
> > **A8**: Our diffusion model (HyperLDM) operates on HVAE latent vectors. The “training data” for our diffusion model consists of fine-tuned HVAE latents, which we compute on the fly, producing each HVAE latent by fine-tuning it on the data from a task sampled from the training task set, using randomized data splits and a randomized epochs number. Due to this procedure, even a limited number of total tasks results in a rich and virtually unlimited “dataset” of fine-tuned HVAE latents used for training the diffusion model. For this reason, and due to our results on the test set, we believe our diffusion model to be working as intended, and not simply overfitting to the training dataset. Another point in favor of the diffusion model working as intended is how our HyperLDM (gamma>=1) results score demonstrably higher compared to the “unconditional” diffusion model with classifier-free gamma=0, which would be equally suited to simply reproduce training samples.
> >
> >
> >
> > Thank you again for your feedback. We hope that the improved clarity and performance will convince the reviewer to vote for an acceptance of the paper. We remain available if you have any further concerns or questions during the remainder of the rebuttal period.

---

> ### Author Response · Authors · 2022-11-18
> **Reminder**
>
> We once again thank you for your review and suggestions. We apologize for the insistence, but we would like to gently remind you that the interactive period will expire in less than a day. We would like to engage in a discussion about any aspects of the paper or of our rebuttal that you may still object to. For this reason, we would like to have the opportunity to read your further replies before our ability to respond is blocked.
> Thanks a lot again, and we apologize for the inconvenience.

---

### Official Review · Reviewer_HRh8 · 2022-10-25

**Confidence:** 4
**Correctness:** 3
**Technical Novelty And Significance:** 3
**Empirical Novelty And Significance:** 4
**Recommendation:** 6

**Clarity, Quality, Novelty And Reproducibility:**

Clarity: This paper is overall well-written, except for that some details are confusing and missing: see weakness 1 and 2.

Quality: The evaluation of the proposed methods are fairly conducted: see strength 2.

Novelty: This paper provide a novel perspective of hypernetwork-based approach: see strength 1.

Reproducibility: This paper provides full code for reproducing the experimental results.

**Strength And Weaknesses:**

Strength:
1. This multi-modal generative modelling of "neural network weight" is novel .
2. The task of Meta-VQA is firstly proposed in this paper, and it is very interesting for me.
3. This paper clearly benchmark the given task including convincing baselines and the proposed method, which largely contributes to (multi-modal) meta-learning community.

Weakness:
1. The proposed method consists of three components, and their usage is still confusing for me (they are used in two ways: w or w/o Hyper-LDM). What I understand is: a) $z_0$ is firstly sampled, e.g. $z_0\sim N(0,I)$, b) $z_0\rightarrow z_t$: the code is iteratively modified through either gradient descent on equation 4 (w/o Hyper-LDM) or LDM, and c) Hyper-Decoder decodes $z_t$ to obtain the weight. If this is not wrong, I would like to recommend either algorithm tables or figure explaining how they are used in the meta-test (inference time).
2. I can't find any information about how exactly the base model perform each task. Does the base model find a word from vocabulary by measuring the cosine distance? Or is it a classifier? Then, how the generated weights are used for the tasks? These questions can not be resolved from this paper.



**Summary Of The Paper:**

- This paper proposes generative models that predicts (or generates) high-performing task-specific weights conditioned on text description of the task.
- Specifically, three "Hyper"-components are proposed:
1) Hyper-"decoder": a decoder of VAE trained with (unconditional) auto-encoding of weight vector,
2) Hyper-"CLIP": task and text encoders trained to maximize cosine similarity of positive pair,
3) (optional) Hyper-"LDM": diffusion model for the latent space of 1) Hyper-"decoder".
- After training the above three components, we can sample task-specific weights through:
 $t \xrightarrow{\text{HyperCLIP}} e \xrightarrow{\text{HyperLDM}} z \xrightarrow{\text{HyperDecoder}} w$.
- The experimental results show that the proposed method performs better than the relevant baselines.




**Summary Of The Review:**

This paper proposes multi-modal generative models learn to generate ``neural network weights'' conditioned on text description on new tasks, and provides a kind benchmark to this research direction. Some of details can be improved.

---

> ### Author Response · Authors · 2022-11-11
> **Response to Reviewer HRh8**
>
> We thank the reviewer for carefully evaluating our work and we are pleased with the overall positive feedback. We particularly appreciated the reviewer’s interest in the Meta-VQA dataset, and in highlighting the multi-modal generative aspect of our work. We will try to address the reviewer’s concerns below point-by-point and hope that we can lift some of the criticism.
>
>
> **P1**:
> > I would like to recommend either algorithm tables or figure explaining how they are used in the meta-test (inference time).
>
> **A1**: We can confirm that your understanding of our method is correct. Indeed, our methods are based on a) sampling z0, b) iteratively updating z0→zt with either gradient descent on equation 4 (using HyperCLIP, not using HyperLDM) or with LDM (without HyperCLIP), c) decoding zt and obtaining model weights.
> To increase clarity, we included algorithm tables in Appendix A.4 and A.5 explaining in detail the train-time and inference-time steps taken in the training of the baselines and of our methods.
>
>
> **P2**:
> > I can't find any information about how exactly the base model perform each task. Does the base model find a word from vocabulary by measuring the cosine distance? Or is it a classifier? Then, how the generated weights are used for the tasks?
>
> **A2**: Our choice of base model is detailed in Appendix A.2, in the “Base Network” chapter. This CLIP-Adapter model, as described in the cited literature, performs classification with a softmax operation over the vector of cosine distances between the image’s CLIP-adapted embedding and the language labels’ CLIP embeddings.
> Our method produces neural network weights for the Adapter component of the CLIP-Adapter base model.
>
> We again thank the reviewer for the constructive feedback, and hope we adequately addressed the reviewer’s questions. We hope our clarifications, together with the additional empirical result in the general response will be convincing for further raising the score. We are happy to discuss any additional points during the remainder of the rebuttal period.

---

> ### Author Response · Authors · 2022-11-18
> **Reminder**
>
> We once again thank you for your review and suggestions. We apologize for the insistence, but we would like to gently remind you that the interactive period will expire in less than a day. We would like to engage in a discussion about any aspects of the paper or of our rebuttal that you may still object to. For this reason, we would like to have the opportunity to read your further replies before our ability to respond is blocked.
> Thanks a lot again, and we apologize for the inconvenience.

---

### Official Review · Reviewer_TMAm · 2022-11-04

**Confidence:** 4
**Correctness:** 2
**Technical Novelty And Significance:** 3
**Empirical Novelty And Significance:** 2
**Recommendation:** 3

**Clarity, Quality, Novelty And Reproducibility:**

The technical background relevant to the method -- hyper-networks, generative hyper-networks, guidance with a contrastive classifier, as well as classifier-free guidance -- is clearly written. A key issue of clarity is the positioning of the work as pertaining to meta-learning -- which involves both task inference and subsequent adaptation -- since the task inference problem is not present in their train and test settings. This is because the model is given the corresponding task description as input, such that the setting reduces to conditional multi-task learning. Moreover, while the methods proposed -- namely, approaches for modulating a base model with hypernetwork layers guided with a task description -- are fairly original, such approaches contribute considerable added complexity without demonstrating clear advantages to the straight-forward baseline of conditional multi-task learning.

**Strength And Weaknesses:**

Strengths:

The authors propose a novel approach for modulating a base model in a task specific manner by guiding the prediction of dynamic weights with a learned representation of the task description.

The authors propose a novel approach for how said task representations can be learned by contrastive learning for joint embedding of task description and dynamic weights.

The technical background relevant to the method -- hyper-networks, generative hyper-networks, guidance with a contrastive classifier, as well as classifier-free guidance -- is clearly written.

Weaknesses:

The experimental setting used to validate the effectiveness of the approach does not necessarily require meta-learning (task inference followed by adaptation), since the model is conditioned on an explicitly provided task description. The work thus seems incorrectly positioned, insofar as it relates more to multi-task learning rather than meta-learning. This results in a lack of clarity, which could be resolved by positioning the approach as a solution for multi-task learning.

The proposed method is not shown to outperform the straight-forward yet strong baseline of conditional multi-task learning, despite being significantly more complex. More experiments are needed that clearly demonstrate the advantage of introducing such complexity.

Popular relevant approaches such as FiLM (Perez et al, 2017)  and AdaIN (Huang et al, 2017) are not discussed, but would serve as strong baselines for base model modulation with a conditioning signal.

**Summary Of The Paper:**

This paper proposes an approach for adapting a base model (in their case, a VQA model) given a task descriptor. The authors propose to perform such adaptation via a hypernetwork layer, and learn a latent representation through which one can control the modulation of the base model by training a VAE on said layer’s weights. This manipulation is achieved by either 1) using a task descriptor learned through joint embedding of task description-parameter pairs for classifier-like guidance in the hypernetwork latent space or 2), fitting a generative model on task descriptor-parameter pairs and using classifier-free guidance on the hypernetwork latent. They demonstrate that these techniques allow them to condition the base model on in-distribution test task descriptions so as to achieve a multi-task learning effect. While the methods considered are somewhat novel and interesting, their merit is not clearly demonstrated experimentally, as it does not clearly outperform the straight-forward yet strong baseline of conditional multi-task learning. Moreover, the setting considered in experiments may not exactly require meta-learning, since the task inference problem is not present at test-time (task description is given), so that the problem reduces to conventional multi-task learning.


**Summary Of The Review:**

While the methods proposed are fairly original, such approaches contribute considerable added complexity without demonstrating clear advantages to the straight-forward baseline of conditional multi-task learning. Perhaps more importantly, a key issue of clarity is the positioning of the work as pertaining to meta-learning -- which involves both task inference and subsequent adaptation -- since the task inference problem is not present in their train and test settings.

---

> ### Author Response · Authors · 2022-11-11
> **Response to Reviewer TMAm**
>
> We thank you for your helpful review, which allowed us to identify weak points in the presentation of our paper’s findings. We would like to separately address the main points you put forth:
>
> **P1**:
> > The proposed method is not shown to outperform the straight-forward yet strong baseline of conditional multi-task learning, despite being significantly more complex. More experiments are needed that clearly demonstrate the advantage of introducing such complexity.
>
> **A1**:  We point to our general response for the clarification on the strength of our method, as well as our new experimental results. In fact, in our experiment where 50% of task descriptors are removed, our method achieves 54.84% **zero-shot accuracy** on unseen tasks, whereas the multitask baselines only achieve 51.68% (Table 1). This improvement in performance is all the more significant when taking into account that MAML only achieves 60.24% accuracy on the same set **after using few-shot examples** (which our method does not use). We hope the reviewer can agree that this result alone is already empirically significant.
>
>
> **P2**:
> > Popular relevant approaches such as FiLM (Perez et al, 2017) and AdaIN (Huang et al, 2017) are not discussed, but would serve as strong baselines for base model modulation with a conditioning signal.
>
> **A2**: We thank you for the suggestion, indeed FiLM would constitute a good comparison, so we included it as a baseline, as a variation of the Cond. Multitask approach. It appears that, at least in our setting, using FiLM as a conditioning method does not lead to improved performance (53.84% vs 54.12%). We did not use AdaIN as another baseline, as the AdaIN technique is specifically designed to work with convolutional networks, while our base model is not a convolutional network (it is a CLIP-Adapter, a sequence of fully connected adapter layers).
>
>
> **P3**:
> > The experimental setting used to validate the effectiveness of the approach does not necessarily require meta-learning (task inference followed by adaptation), since the model is conditioned on an explicitly provided task description. The work thus seems incorrectly positioned, insofar as it relates more to multi-task learning rather than meta-learning. This results in a lack of clarity, which could be resolved by positioning the approach as a solution for multi-task learning.
>
> **A3**: Thank you for this comment. We believe you raise a very important point on the clarity of the positioning of our work. As detailed in the General Response point 2, the inclusion of few-shot results in the main table gave the wrong impression that our main experimental setting necessarily requires meta-learning. Indeed, in practice our method could be classified as a multi-task learning method, due to its zero-shot nature.
> We however feel that there are several reasons for why using the meta-learning terminology is relevant to how we approached the problem. Our main argument for this is that, as described in the Introduction and in Section 4.1, we attempt to reframe learning (MAML-like) network adaptations as a multi-modal learning problem. We consider "few-shot examples" and "a natural language description" to be just two possible modalities that can be used to conditionally adapt a neural network for a novel task. Then, HyperCLIP guidance or HyperLDM can be considered as "simulating" a few-shot learning process, using a language instruction in lieu of actual data from the task to perform both task inference and adaptation in our HVAE latent space. For this reason, we believe that the connection with "meta-learning" (as "learning how to do better adaptations") is appropriate. We are in any case open to further discussion on this point.
>
>
> We again thank the reviewer for the constructive feedback, and hope we adequately addressed the reviewer’s questions. We hope our clarifications, together with the additional empirical result in the general response will be convincing for voting for an acceptance of the paper. We are happy to discuss any additional points during the remainder of the rebuttal period.

---

> > ### Comment · Reviewer_TMAm · 2022-11-17
> > **Clarification on the Meaning of Zero-shot**
> >
> > Could you please clarify how the training and test sets of the Meta-VQA dataset are constructed? My understanding is that the term "zero-shot" applies to settings where the test task distribution is different from that of the training task distribution. From what I can tell, the questions considered in the Meta-VQA train and test splits come from the same distribution. If that is so, I am not sure this should be considered a zero-shot setting, since the model is trained on problems from the same distribution as those seen at test-time.

---

> > > ### Author Response · Authors · 2022-11-18
> > > **Re: Clarification on the Meaning of Zero-shot**
> > >
> > > Thank you for your reply and further question. The Meta-VQA dataset’s training and test sets are constructed by performing a random split of questions (tasks), which are non-overlapping (no identical or semantically identical questions are present in both sets). This indeed doesn’t mean that the training and test task distributions are necessarily different, as all tasks can be considered as having been sampled from the same “Meta-VQA” task distribution. A task can be “unseen” even if it doesn’t ultimately come from a different task distribution. Similarly, what is meant by “generalization” in machine learning doesn’t imply by default distribution shift.
> > > We believe that the common definition of “zero-shot” learning is completely independent from the concept of “out of distribution generalization”. Out of distribution generalization lies outside of the scope of our work.
> > >
> > > By zero-shot learning, we mean that we are able to perform adaptation on an unseen task without being exposed to examples to learn the task in the inner loop, and only using the natural language description/question. This is in contrast to few-shot learning, in which we possess examples (a support set) to learn the task. To give an example of our use of “zero-shot learning” in the literature, we point to the notorious “Language models are few-shot learners” paper [1], specifically to the footnote in page 4. We report here an excerpt:
> > > > [...], we use the term “meta-learning” to capture the inner-loop / outer-loop structure of the general method, and the term “in context-learning” to refer to the inner loop of meta-learning. We further specialize the description to “zero-shot”, “one-shot”, or “few-shot” depending on how many demonstrations are provided at inference time. These terms are intended to remain agnostic on the question of whether the model learns new tasks from scratch at inference time or simply recognizes patterns seen during training – this is an important issue which we discuss later in the paper, but “meta-learning” is intended to encompass both possibilities, and simply describes the inner-outer loop structure.
> > >
> > > We believe that the above quote neatly describes our setting as well. Moreover, it is also an example of a usage of “meta-learning” in a way that is compatible with our method, to further address your previous concern on the positioning of our work.
> > >
> > > [1] Brown, Tom, et al. "Language models are few-shot learners." Advances in neural information processing systems 33 (2020): 1877-1901. https://arxiv.org/abs/2005.14165

---

### Author Response · Authors · 2022-11-11
**General Response to Reviewers (1/2)**

We thank all the reviewers for their detailed feedback as well as their constructive comments on the paper. The review made us realize a crucial lack of clarity in our original paper, which we worked hard on addressing. Here, we want to draw attention to the following points and new results which we believe strengthen the paper in several aspects.

## 1) Improved presentation of the strengths of our approach

A common criticism from reviewers is that our experimental results do not appear to convincingly beat the straightforward multi-task baseline. Thus, the additional complexity required by our method would appear unjustified. We recognize that we did not present our results optimally and would like to elaborate on this.

As is the case for the image synthesis setting, neatly separating an unconditional generative model from a “guidance” component allows us to leverage **unconditional pre-training** - that is, training the generative model on tasks that are not labeled by a task descriptor. This property is attractive when obtaining meaningful task description may be expensive. For instance, one can think of a behavioral cloning setting, where obtaining meaningful diverse demonstration trajectories (tasks) might be easy while labeling them with the corresponding semantic not as much. Our approach can leverage tasks without textual descriptors during the learning of the unconditional Hypernetwork-VAE (HVAE), and use a smaller number of tasks with textual descriptors to learn the guidance component (HyperCLIP/HyperLDM) on top.

To simulate this setting, we ran an experiment on the Meta-VQA dataset by removing the task descriptor for a percentage of tasks from the dataset. The figure related to this experiment was already present in our original submission, but unfortunately was not clearly highlighted and was lacking summary statistics.

We now highlight results from this experiment in the main results table in the manuscript (Table 1, also reported below). As expected, the conditional multi-task learning baselines fail here, even overfitting compared to the unconditional baseline due to the relative lack of conditional data. Our approach on the other hand barely loses performance compared to the setting in which conditioning descriptors are fully available for all training tasks.  We think this setting convincingly demonstrates the strength of our method.

## 2) Removed few-shot performance from the main table

We also realize that the inclusion of the few-shot performances in the main table may have been misleading for the positioning of our work. Indeed, our method’s main setting is that of zero-shot learning given a new task. We included the few-shot performance to provide an informal performance upper bound for our method with our fixed choice of base model. Indeed, the full knowledge of the task, namely the task descriptor as well as few shot data, allows the test set accuracy to barely surpass **60%** - a highly relevant context for judging the performance gain of our algorithm in the zero shot setting. However, we realize that providing the few-shot adaptation performance from all methods in the main table has been more confusing than informative.

To focus on zero-shot performance, we now have removed the few-shot numbers from the main table, and have updated Section 6.2 with further discussion reflecting the above point instead. The few-shot numbers can still be found in Appendix A.6.

---

> ### Author Response · Authors · 2022-11-11
> **General Response to Reviewers (2/2)**
>
> ## 3) Improved general performance
>
> We further improved the performance of our HyperLDM algorithm during the rebuttal period. We have found that training a generative model on a richer set of fine-tuned models improved the performance of the subsequent guidance generation. We describe the training steps in detail in Appendix A.5, clarifying the process for both this new run of experiments and for the old experiments: the technique used for the results shown in our original submission corresponds to using a HVAE with “Variant 1” training, while the new results correspond to “Variant 2” (See Appendix A.5.1).
> As you can see in the table below, our overall performance further increased, making our contribution score better in all scenarios compared to the strongest multi-task baseline. While the performance improvement seems to remain small, **it is non-negligible when taking into account the performance upper bound of around 60% given by the few-shot adaptation, and it is also statistically significant**.
>
> | Method                    	| Zero-shot (50% Q.)    	| Zero-Shot (100% Q.)   	|
> |-------------------------------|---------------------------|---------------------------|
> | Cond. Multitask           	| 51.68 (±0.33)         	| 54.12 (±0.80)         	|
> | Cond. HNet-MAML           	| 51.54 (±0.63)         	| 53.02 (±0.20)         	|
> | Cond. Multitask FiLM      	| 51.60 (±0.56)         	| 53.84 (±0.61)         	|
> |-------------------------------|---------------------------|---------------------------|
> | HNet + HyperCLIP Guidance   | 53.51 (±0.22)         	| 53.98 (±0.54)         	|
> | HVAE + HyperCLIP Guidance   | 53.82 (±0.07)         	| 53.91 (±0.08)         	|
> | HNet + HyperLDM $\gamma$=1  | 53.66 (±0.25)         	| 54.06 (±0.21)         	|
> | HNet + HyperLDM $\gamma$=1.5| 54.08 (±0.11)         	| 54.30 (±0.27)         	|
> | HVAE + HyperLDM $\gamma$=1  | 54.72 (±0.23)         	| 55.03 (±0.32)         	|
> | HVAE + HyperLDM $\gamma$=1.5| **54.84 (±0.24)**     	| **55.10 (±0.08)**     |
>
>
> We again thank all the reviewers for their help in highlighting key improvement points for our paper. We hope the suggested changes will address them, as well as adequately convey the story we intended to tell, and will convince the reviewers to raise their scores.

---

### Author Response · Authors · 2022-11-16
**Update: Rebuttal time running out**

Thank you all once again for your useful comments. We believe we have addressed many of the reviewers' concerns - yet, as the end of the rebuttal period is coming close, we see that you have not yet posted replies or updated your reviews in response to our new results and changes. In case something is unclear, please contact us with sufficient time before the discussion period ends.

---

### Decision · Program_Chairs · 2023-01-20

**Decision:**

Reject

**Justification For Why Not Higher Score:**

N/A

**Justification For Why Not Lower Score:**

N/A

**Metareview: Summary, Strengths And Weaknesses:**

This paper proposed a meta-learning method that uses classifier(-free) guidance from the generative modeling literature to generate zero-shot adapted network weights. While the paper contains some interesting ideas, the majority of the reviewers generally agreed that the work is not strong enough for acceptance (especially the results of the initial manuscript are rather poor which may explain the low rating review scores). Reviewers raised some major weakness concerns about incremental empirical improvement as compared with the simple baselines, the weak experiments, the lack of convincing results to justify the added complexity of the method, and the confusing presentation issues.

While we appreciate the authors have tried to improve the paper and add additional experimental results to address the review concerns during the rebuttal, the major concerns of the reviews remain and most of the reviewers were still not convinced that the revision is strong enough for acceptance (while some reviewer did acknowledge the efforts of adding additional results help improve the quality somewhat but was not convinced to change the reject decision).

Overall, the quality of this work is below the acceptance bar and the paper would require significantly more revision efforts to improve the overall quality, which includes more extensive experiments, more convincing results to justify the significance of the proposed method with the added complexity, and better presentation and writing to improve the motivation of the proposed technique and the overall clarity of the work.